# Diagnostic Efficacy of Serum Asialo α1-Acid Glycoprotein Levels for Advanced Liver Fibrosis and Cirrhosis in Patients with Chronic Hepatitis B Compared to That in Healthy Subjects: A Prospective Study

**DOI:** 10.3390/jcm12020712

**Published:** 2023-01-16

**Authors:** Yoonseok Lee, Seryun Bae, Ji Hoon Kim, Minjung Kwak, So Yeon Jeon, Taehyung Kim, Sun Young Yim, Young-Sun Lee, Young Kul Jung, Yeon Seok Seo, Hyung Joon Yim, Jong Eun Yeon, Kwan Soo Byun

**Affiliations:** 1Department of Internal Medicine, Division of Gastroenterology and Hepatology, Korea University College of Medicine, Seoul 08308, Republic of Korea; 2Department Digital Information and Statistics, Pyeongtak University, Pyeongtaek-si 17869, Republic of Korea

**Keywords:** liver cirrhosis, liver fibrosis, biomarker, chronic hepatitis B

## Abstract

Background: Serum asialo α1-acid gycoprotein (AsAGP) is a novel biomarker specific to liver fibrosis. Aim: To evaluate the diagnostic efficacy of serum AsAGP levels in classifying the severity of liver fibrosis and differentiating liver cirrhosis (LC) in patients with chronic hepatitis B (CHB) from healthy controls. Methods: Overall, 206 subjects were prospectively enrolled. LC was diagnosed based on liver stiffness levels (>11 kPa) measured using transient elastography. Serum AsAGP levels were measured using an antibody-lectin sandwich immunoassay. We investigated the diagnostic performance by comparing serum AsAGP levels among healthy control, CHB, and CHB with LC groups. Sensitivity, specificity, and optimal AsAGP cut-off values were also calculated. Results: Serum AsAGP levels were significantly different between healthy controls, CHB patients, and CHB patients with LC (1.04 ± 0.31 µg/mL, 1.12 ± 0.34 µg/mL, 1.51 ± 0.43 µg/mL respectively; *p* < 0.001). Serum AsAGP levels positively correlated with liver stiffness (r = 0.46, *p* < 0.001). AUROC of healthy control versus CHB with LC was 0.821 (*p* < 0.001, optimal cut-off 1.036 µg/mL). AUROC of healthy control versus CHB was 0.624 (*p* = 0.049, optimal cut-off level 0.934 µg/mL). AUROC of CHB versus CHB with LC was 0.765, (*p* < 0.001, optimal cut-off 1.260 µg/mL). Conclusions: Serum AsAGP levels in CHB patients with LC were significantly higher than those in healthy controls and CHB patients. AsAGP levels showed good diagnostic performance in predicting advanced fibrosis and cirrhosis, which suggests a potential role as a biomarker for predicting the progression of liver disease in CHB.

## 1. Introduction

Hepatitis B virus (HBV) infection is a major chronic liver disease, with a global prevalence of 3.5% in 2015; more than 600,000 patients die due to HBV-associated disease annually [1,2]. Liver fibrosis is a main prognostic factor, and the severity of liver fibrosis and liver cirrhosis (LC) correlates with the development of hepatic decompensation and hepatocellular carcinoma (HCC) [3,4]. Therefore, early detection of liver fibrosis and prevention of fibrosis progression are important for the management of patients with chronic hepatitis B (CHB) [5].

Almost all chronic liver diseases, including viral infections, alcoholic liver disease, and nonalcoholic fatty liver disease (NAFLD), result in liver fibrosis. Chronic liver injury leads to the excessive accumulation of extracellular matrix proteins and destroys the hepatic architecture. Consequently, liver fibrosis progresses to LC in the end-stage, regardless of aetiology [6]. Discriminating asymptomatic liver fibrosis and cirrhosis in chronic liver disease is vital for successful management [7].

Although liver biopsy is the gold standard for evaluating liver fibrosis and cirrhosis [8], it is an imperfect method. Its limitations include high cost, sampling variability due to heterogeneity of liver fibrosis, inter- and intra-observer variability [9]. In addition, liver biopsy has potential risk of morbidity and mortality associated with invasive procedures [10]. Hence, non-invasive tests for assessment of liver fibrosis have been developed, such as serologic biomarkers, transient elastography (TE), acoustic radiation force impulse imaging, and magnetic resonance elastography (MRE) [11]. Among these, TE is widely performed instead of liver biopsy in clinical practice. TE predicts liver fibrosis based on liver stiffness measurement and demonstrates reliable performance in predicting fibrosis and cirrhosis [7,12,13]. However, it requires expensive device, extra space, and dedicated personnel for measurement. Among the biomarkers for assessing fibrosis, the FIB-4 index, based on laboratory parameters, is valuable in that it can be easily performed in clinical practice. Its performance for assessing liver fibrosis has been proven in patients with CHB and chronic hepatitis C [14,15]. However, the accuracy of the FIB-4 index for assessing liver fibrosis is not sufficient to replace liver biopsy [16,17]. Therefore, simpler and more reliable biomarkers are needed.

α1-acid glycoprotein (AGP) is an acute phase protein that is mainly synthesised and cleared by the liver [18]. Removal of terminal sialic acid of AGP during clearance results in the formation of asialo α1-acid glycoprotein (AsAGP). Asialo glycoprotein receptors of hepatocytes capture circulating AsAGP and remove it from the serum [19]. Therefore, serum AsAGP concentration can be affected by liver diseases. It has been reported that AsAGP levels increase in patients with liver diseases including LC and HCC [20,21,22]. Recently, two studies evaluated the diagnostic accuracy of serum AsAGP levels to predict the severity of liver fibrosis and cirrhosis [23,24], and AsAGP showed better diagnostic performance in patients with CHB than in patients with nonalcoholic fatty liver disease [24]. These studies suggested the potential of AsAGP as a serum biomarker of fibrosis, but no prospective study has compared it with healthy controls. Hence, we prospectively investigated the diagnostic accuracy of serum AsAGP levels for assessing presence and degree of fibrosis and diagnosing cirrhosis in CHB patients compared with that in healthy controls.

## 2. Materials and Methods

### 2.1. Study Design and Subjects

This was a single-centre prospective cohort study. We enrolled patients with CHB, including LC patients and healthy controls who visited the Department of Hepatology at Korea University Guro Hospital. All study subjects were Asians of Korean and Chinese nationality. The inclusion criteria for CHB patients were as follows: (1) HBsAg positivity for more than 6 months, (2) age older than 19 years, and (3) ability to undergo TE examination. The exclusion criteria for CHB patients were as follows: (1) alcohol consumption (>30 g/day in men or 20 g/day in women); (2) malignant disease except cure after 5 years; 3) alanine aminotransferase (ALT) > 5 times the upper normal limit; (3) total bilirubin > 2 mg/dL; (4) coinfection of HCV; (5) chronic heart failure; (6) ascites; and (7) pregnancy or breastfeeding. Cases that fulfilled all the following strict criteria were enrolled as healthy controls: (1) patients without CHB or other liver diseases; (2) ALT within normal limits; (3) no diabetes; (4) no hypertension; (5) platelet count of 150,000/mm^3^ or more; (6) no fatty liver on sonography; (7) alcohol consumption less than 30 g/day in men or 20 g/day in women; and (8) did not meet any of the exclusion criteria of the CHB group. Most of the healthy controls were visiting the hospital to check for cyst or hemangioma. Liver ultrasonography and TE were performed at baseline (the first visit after screening). The subjects included in the healthy control group did not undergo liver stiffness measurement because the liver stiffness of healthy controls was outside the scope of this study. Fibrosis stage and cirrhosis in the CHB group were defined based on liver stiffness values measured by TE (cut-off value: ≥11 kPa for cirrhosis, 8.1–10.9 kPa for stage 3 fibrosis, 7.2–8.0 kPa for stage 2 fibrosis, <7.2 kPa for stage 0–1 fibrosis) [25].

All the participants were enrolled after providing their written informed consent for participation. The Institutional Review Board of the Korea University Guro Hospital approved this study (2020GR0096).

### 2.2. Liver Stiffness Assessment

TE (FibroScan^®^, EchoSens, Paris, France) was used as the reference for fibrosis. An experienced operator blinded to the participants’ characteristics performed the test. The median value represents the stiffness of the liver tissue. The results that fulfilled the following criteria were regarded as valid: (1) at least 10 valid measurements; (2) a success rate above 60%; and (3) a ratio of the interquartile range to median < 30%.

### 2.3. Measurement of Serum AsAGP Level

The patient’s blood was collected after an overnight fast of at least 8 hours at baseline. The patient’s blood was centrifuged at 2000 rpm for 10 min in a serum separation tube, and the supernatant was immediately transferred to the laboratory unit for analysis. The serum AsAGP level was measured by an antibody-lectin sandwich immunoassay using an AceGP^®^ ELISA kit (ACEBiomed Inc., Seoul, Korea) [26]. For more accurate results, we performed triplicate analysis and used the average value for serum AsAGP concentration.

### 2.4. Sample Size Calculation

The formula for calculating the number of subjects in this clinical trial is as follows:$$n=\frac{\sigma^{2}{\left(z_{\frac{\alpha }{2}}+z_{\beta }\right)}^{2}}{d^{2}}$$

Here, d is the error range and V(AUROC) =$\;\sigma^{2}=0.0099\times exp\left(-\frac{a^{2}}{2}\right)\times \left(6a^{2}+16\right)$, $a=\varphi^{-1}\left(AUC\right)\times 1.414$.

Based on a previous retrospective study, we assumed that the areas under the receiver operating characteristic curves (AUROC) of serum AsAGP levels for CHB patients and CHB patients with LC compared to those for healthy controls were 0.914 and 0.982, respectively [26]. As the AUROC of CHB patients and healthy controls was 0.914, 63 subjects were needed to estimate a significance level of 5% and an error range of 6%. In addition, the AUC of CHB patients with LC and healthy controls was 0.982; 36 subjects were needed to estimate a significance level of 5% and an error range of 3%. Therefore, this study included 63 participants per group. Considering a dropout rate of 10%, we decided to enroll a minimum of 70 participants per group of healthy controls, CHB patients, and CHB patients with LC.

### 2.5. Statistical Analysis

The primary endpoint was to investigate the sensitivity, specificity, and AUROC in CHB patients and CHB patients with LC compared to healthy controls using AsAGP. The secondary endpoint was to investigate the sensitivity, specificity, and AUROC for each stage of fibrosis using AsAGP. Data were reported as mean (standard deviation) or numbers with percentages. One-way ANOVA and pairwise post-hoc testing were performed to analyse the differences in the serum AsAGP levels between each group. The correlation between serum AsAGP levels and other variables was calculated using the Pearson correlation coefficient. We then computed the AUROC of the serum AsAGP levels for the diagnosis of chronic hepatitis and cirrhosis. The sensitivity, specificity, and optimal cut-off values were obtained from the ROC curves. Univariate and multivariate analyses by stepwise logistic regression were performed to determine whether the serum AsAGP level was a significant predictor for the diagnosis of CHB with LC. All data were analysed using the R software (version 4.0.0; R Foundation for Statistical Computing, Vienna, Austria).

## 3. Result

### 3.1. Baseline Characteristics

We screened 227 participants (77 in healthy controls, 71 in CHB patients, and 79 in CHB patients with LC); however, with 21 screening failures, we ended with a final total of 206 participants (68 healthy controls, 70 CHB patients, 68 CHB patients with LC). All participants were enrolled from April 2020 to December 2020 (Figure 1). Table 1 shows the baseline characteristics of the included participants.

The mean age was 53.3 years without significant differences among the groups. There were 100 males (48.5%) and 106 females (51.5%) in our study. There were more men in the CHB and CHB with LC groups than in the healthy control group. Among the patients with CHB and CHB with LC, 20 (9.7%) had diabetes, which was more prevalent in CHB patients with LC, and 35 (17%) had hypertension. Aspartate aminotransferase (AST) and ALT levels were higher in CHB patients and CHB patients with LC than in healthy controls. Mean liver stiffness of the CHB group was 6.9 ± 2.3 kPa and that of the CHB with LC group was 18.8 ± 7.9 kPa.

### 3.2. Serum AsAGP Level Accroding to Fibrosis Stages

Serum AsAGP levels were significantly higher in the CHB with LC group (1.505 ± 0.432 μg/mL) than in the healthy control (1.036 ± 0.308 μg/mL, *p* < 0.001) and the CHB groups (1.121 ± 0.337 μg/mL, *p* < 0.001) (Table 2, Figure 2).

There was no significant difference in the serum AsAGP levels between the healthy control group and the CHB without cirrhosis group. The distribution of serum AsAGP levels according to the fibrosis stage is summarized in Table 3 and Figure 3.

The fibrosis stage was classified based on liver stiffness measured using TE. Among the 70 CHB patients, 39 had fibrosis stage 0–1, 8 had fibrosis stage 2, and 23 had fibrosis stage 3. Overall, serum AsAGP levels increased as the fibrosis stage increased (*p* < 0.001 by one-way ANOVA). When post-hoc analysis was performed to evaluate whether there was a significant difference in AsAGP levels between each fibrosis stage, the serum AsAGP level in CHB patients with LC was significantly increased compared to that in populations with other fibrosis stages (healthy control, F0–1, F2, and F3). However, the CHB group (F0–1, F2, and F3) did not show significant differences compared to the healthy control group.

### 3.3. Correlation between Serum AsAGP Level and Baseline Variables including Liver Stiffness

Table 4 shows the Pearson’s correlation between serum AsAGP levels and other variables, including liver stiffness, which reflects the status of liver function. Serum AsAGP levels showed a significant positive correlation with AST (r = 0.227, *p* = 0.001) and ALT levels (r = 0.160, *p* = 0.021). Liver stiffness and serum AsAGP levels were also significantly correlated (r = 0.436, *p* < 0.001). In contrast, serum bilirubin and serum albumin, which showed a significant correlation with liver stiffness, did not correlate with AsAGP levels.

### 3.4. Diagnostic Performance of AsAGP Level

We analysed the AUROC, sensitivity, and specificity to evaluate the efficacy of serum AsAGP levels for predicting liver fibrosis and cirrhosis in CHB patients. The AUROCs of serum AsAGP levels for CHB patients and CHB patients with LC compared to those for healthy controls were 0.624 (95% CI 0.528–0.720; *p* = 0.049) and 0.821 (95% CI 0.751–0.890; *p* < 0.0001), respectively (Table 5, Figure 4A,B).

The corresponding optimal cut-off values were 0.934 and 1.036, respectively. The sensitivity and specificity of the optimal cut-off values were 75.7% and 55.9%, respectively, in CHB patients compared to those in healthy controls, and 83.8% and 67.6%, respectively, in CHB patients with LC compared to those in healthy controls. (Table 5) The AUROC for the serum AsAGP level for predicting CHB with LC compared to that for predicting CHB was 0.765 (95% CI 0.682-0.848 *p* < 0.0001, optimal cut-off 1.183 µg/mL) (Figure 3C). The AUROCs of serum AsAGP for predicting each fibrosis stage when comparing healthy controls were 0.610 (95% CI 0.503–0.717; *p* = 0.054) in CHB F0-1, 0.513 (95% CI 0.342–0.684, *p* = 0.087) in CHB F2, and 0.685 (95% CI 0.562–0.809; *p* = 0.063) in CHB F3 (Appendix A). The AUROCs of serum AsAGP for predicting advanced fibrosis (≥F3) when comparing healthy controls was 0.775 (95% CI 0.711–0.840; *p* = 0.033, optimal cut-off 1.288 μg/mL, sensitivity 59.3%, specificity 87.8%) (Appendix A).

### 3.5. Logistic Regression Analyses of Predictors for Detecting CHB with LC

In comparison with healthy controls and CHB patients with LC, age, male sex, AST, ALT, gamma-glutamyl transferase (GGT), albumin, platelet, and AsAGP levels were significantly different in the univariate logistic regression analysis. We used stepwise logistic regression to select significant predictors. Among them, ALT, albumin levels, platelet count, and AsAGP levels (odds ratio 54.514, *p* < 0.001) were significant predictors of CHB with LC. In the comparison of CHB and CHB with LC groups, diabetes and AST, ALT, GGT, total bilirubin, albumin, platelet, and AsAGP levels were significantly different in the univariate analysis. Among them, GGT, platelet count and AsAGP levels (odds ratio 11.898, *p* < 0.001) were independent predictors of CHB with LC in multivariate analysis (Table 6).

## 4. Discussion

In this prospective study, we revealed the clinical usefulness of AsAGP in diagnosing advanced fibrosis and cirrhosis in CHB patients. Serum AsAGP levels showed excellent diagnostic performance (AUROC 0.821, *p* < 0.001) for CHB patients with LC compared to those with healthy controls. Serum AsAGP levels also showed good performance in discriminating CHB patients with LC from CHB patients without cirrhosis (AUROC 0.765, *p* < 0.001). Moreover, serum AsAGP levels showed reasonable discriminating ability in CHB patients without cirrhosis compared with those in healthy controls (AUROC 0.624, *p* < 0.001). To our knowledge, this is the first prospective study to investigate the diagnostic performance of AsAGP in CHB patients compared with that in healthy controls.

Given that presence and degree of liver fibrosis are critical prognostic factor for hepatic complications and HCC in CHB, early detection and repeated assessment of fibrosis progression are needed for individualized management of CHB [27,28]. In the assessment of liver fibrosis, liver biopsy has been regarded as the reference method. However, there are limitations mentioned above. To overcome theses limitations, non-invasive methods including serum biomarkers have been developed. However, currently used biomarkers such as ARPI and FIB-4 are not only validated mainly in hepatitis C and NAFLD but also have low accuracy in diagnosing intermediate-stage fibrosis [29]. In the case of CHB, TE is the most widely used and validated method, but it requires costly devices and the accuracy is suboptimal in patients with obesity and elevated ALT [29,30]. Hence, it is necessary to develop novel biomarkers that can accurately diagnose fibrosis and cirrhosis. Previous studies reported that serum AsAGP correlates with various liver diseases, such as alcoholic liver disease, chronic hepatitis, LC, and HCC [31,32,33,34,35,36]. Recently, an ELISA kit for AsAGP that is simple to perform and can be easily repeated has been developed [37].

Kim et al. [26] evaluated the clinical efficacy of serum AsAGP in 610 patients with heterogeneous liver diseases, 42 healthy controls, and 155 patients with non-hepatic diseases and concluded that serum AsAGP was a promising biomarker for assessing liver disease. However, it was a retrospective study with stored samples and used both healthy participants and patients with non-hepatic diseases as a reference for healthy controls. Our prospective study included 68 healthy controls satisfying the strict criteria mentioned earlier. Lim et al. [23] investigated the performance of serum AsAGP in 48 patients with chronic hepatitis and 48 patients with LC. LC was defined as MRE ≥ 5 kPa. This study concluded that serum AsAGP is a suitable method for differentiating cirrhosis from chronic hepatitis. However, their study did not estimate the sample size and included various heterogeneous etiologies of liver disease; the majority was NAFLD (54.2%). Kim et al. [24] evaluated the performance of serum AsAGP levels for liver fibrosis or cirrhosis in 48 CHB and 75 NAFLD patients. They concluded that the AsAGP level demonstrated reasonable performance in predicting advanced fibrosis or cirrhosis in CHB patients but not in NAFLD patients. Their study estimated a sample size of 105 each for CHB and NAFLD patients, but they were unable to meet this estimate. This study was conducted as planned, overcoming the limitations of previous studies. We sufficiently calculated the sample size and included healthy controls using strict criteria (most of them were health checkups, simple cysts, or hemangiomas), as well as homogenous CHB and CHB with LC patients.

Although our study determined the level of serum AsAGP for differentiating CHB patients from healthy controls and CHB with LC patients from CHB patients or healthy controls, there was no significance in serum AsAGP levels for differentiating CHB patients with F0-1, F2, and F3 fibrosis from healthy controls. (Appendix A) However, our study did not plan to evaluate the differentiation between fibrosis stages. Therefore, the number of patients in each fibrosis stage was not sufficient for optimal analysis, especially for CHB F2 (only eight patients). Early-stage fibrosis (≤F2) could not be reliably divided through TE, MRE, and other serum biomarkers [38]. Further studies with an increased number of participants with early-stage fibrosis are needed to elucidate the performance of AsAGP in fibrosis. Nevertheless, serum AsAGP showed significant diagnostic performance when limited to CHB patients with advanced fibrosis (≥F3) (Appendix A). This result also implies that serum AsAGP can be used as reliable biomarker for detecting advanced fibrosis in CHB patients.

This study has several limitations. First, we defined the stages of liver fibrosis and cirrhosis based on liver stiffness, but not liver biopsy or MRE. To date, liver biopsy has been used as a reference method. Additionally, MRE is more accurate than liver stiffness measurement using TE. However, liver biopsy is limited due to its invasive nature, and MRE performance in assessing fibrosis and cirrhosis has been reported to be comparable with TE [39]. Further studies using liver biopsy or MRE as references are needed. Second, the design of this study was single-centre cross-sectional. Accordingly, it is uncertain whether serial assessment of AsAGP levels would enable tracking fibrotic burden and predicting clinical outcomes, such as liver-related morbidity and liver-related death. Further large, multi-centre, longitudinal studies are needed. Third, although previous studies [23,24] have reported no significant correlation, ALT levels were positively correlated with serum AsAGP levels in this study. One study [23] with heterogenous patients had higher ALT levels than this study, which might have skewed the correlation between ALT and AsAGP. Another study [24] showed similar ALT levels in CHB patients in this study but did not present a correlation between ALT and serum AsAGP levels among CHB patients only. Moreover, only 22.9% of CHB patients were taking antivirals. Because we mainly enrolled CHB patients maintained on antivirals for years (89.3% of all CHB patients) or with an inactive immune state to avoid overestimation of liver stiffness by TE, only nine patients (three in CHB and six in CHB with LC) had ALT levels greater than the upper normal limit (maximum 95 IU/L). However, further research is required to elucidate the relationship between ALT and AsAGP levels. Fourth, TE was not performed in the healthy control group according to the study design. Therefore, the relationship between AsAGP levels and TE levels could not be evaluated in healthy controls. Fifth, there were more women in the healthy control group because most of the healthy controls were visiting the hospital to check for cyst or hemangioma, which are more prevalent in females.

In conclusion, serum AsAGP levels showed good performance in differentiating CHB patients and CHB patients with LC from healthy controls. In addition, the serum AsAGP levels showed promise in predicting advanced fibrosis in CHB patients compared to that in healthy controls. Therefore, serum AsAGP levels can be used as potential biomarkers for advanced liver fibrosis and cirrhosis.

## Figures and Tables

**Figure 1 jcm-12-00712-f001:**
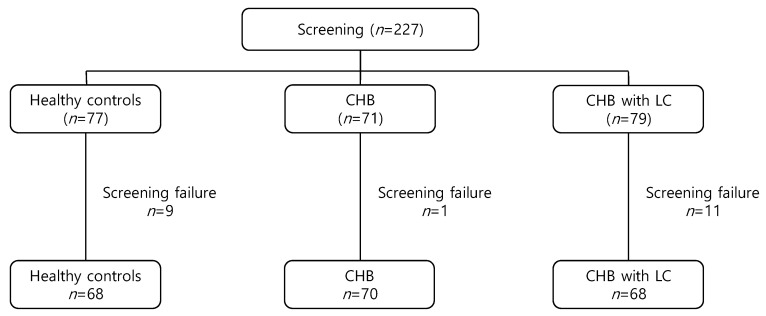
The diagram of patient disposition. CHB, chronic hepatitis B; LC, liver cirrhosis.

**Figure 2 jcm-12-00712-f002:**
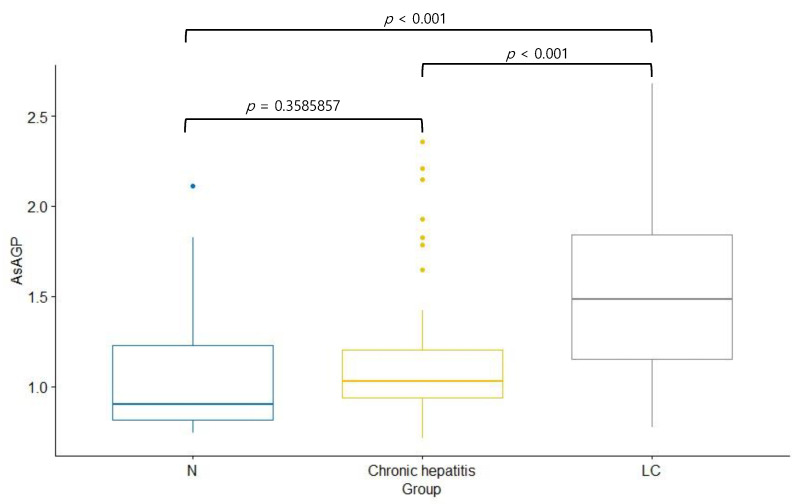
The comparison of AsAGP level according to respective groups. AsAGP, asialo α1-acid glycoprotein; CHB, chronic hepatitis B; LC, liver cirrhosis.

**Figure 3 jcm-12-00712-f003:**
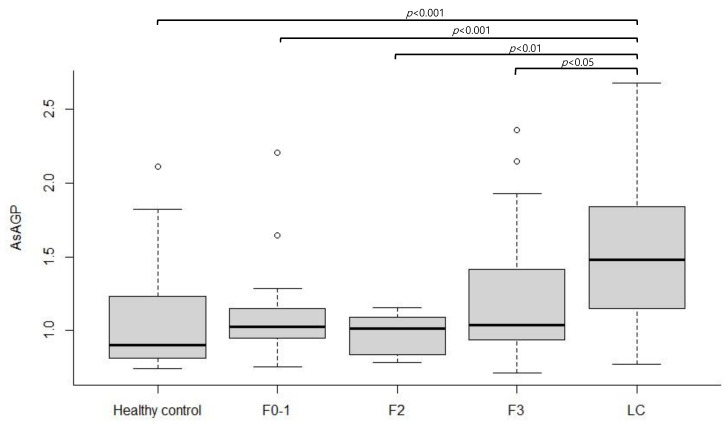
The comparison of AsAGP levels according to respective fibrosis stages. AsAGP, asialo α1-acid glycoprotein; F, fibrosis stage; LC, liver cirrhosis.

**Figure 4 jcm-12-00712-f004:**
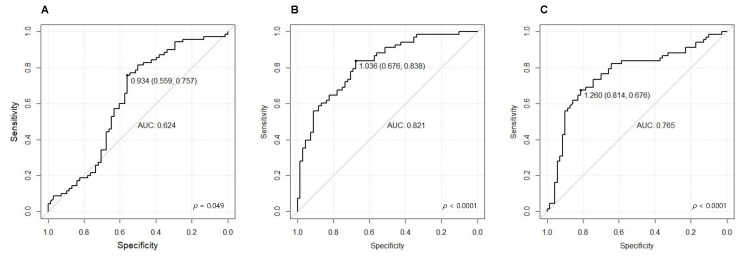
The AUROC of serum AsAGP levels for differentiating liver fibrosis and cirrhosis in CHB patients compared to that in healthy controls. (**A**) CHB patients versus healthy controls, (**B**) CHB patients with LC versus healthy controls, (**C**) CHB patients versus CHB patients with LC.

**Table 1 jcm-12-00712-t001:** Baseline characteristics of study participants.

Variable	Total (*n* = 206)	Healthy Controls (*n* = 68)	CHB Patients (*n* = 70)	CHB Patients with LC (*n* = 68)	*p*-Value
Age (years)	53.3 ± 11.3	50.9 ± 13.4	53.7 ± 10.4	55.3 ± 9.6	0.07
Male gender (number, %)	100 (48.5)	18 (26.5)	43 (61.4)	39 (57.4)	<0.001
Diabetes (number, %)	20 (9.7)	0 (0.0)	6 (8.6)	14 (20.6)	<0.001
Hypertension (number, %)	35 (17.0)	0 (0.0)	17 (24.3)	18 (26.5)	0.001
AST (IU/L)	30.1 ± 14.4	21.9 ± 6.4	29.6 ± 14.2	38.8 ± 15.4	<0.001
ALT (IU/L)	28.8 ± 18.8	18.1 ± 8.3	29.8 ± 17.8	38.5 ± 21.7	<0.001
Total bilirubin (mg/dL)	0.9 ± 1.0	0.8 ± 1.5	0.9 ± 0.4	1.1 ± 0.5	0.309
Albumin (g/dL)	4.2 ± 0.3	4.3 ± 0.2	4.3 ± 0.2	4.1 ± 0.3	0.01
Liver stiffness (kPa)	12.8 ± 8.3 *	NA	6.9 ± 2.3	18.8 ± 7.9	<0.001

* Values of CHB and CHB with LC groups (*n* = 138). ALT, alanine transaminase; AST, aspartate transaminase; CHB, chronic hepatitis B; LC, liver cirrhosis.

**Table 2 jcm-12-00712-t002:** Serum AsAGP levels in respective groups.

Participants Group	Total (*n* = 206)		*p*-Value
	*n*	AsAGP (μg/mL)	
Healthy controls	68	1.036 ± 0.308	<0.001
CHB patients	70	1.121 ± 0.337	
CHB patients with LC	68	1.505 ± 0.432	

AsAGP, asialo α1-acid glycoprotein; CHB, chronic hepatitis B; LC, liver cirrhosis.

**Table 3 jcm-12-00712-t003:** Serum AsAGP levels according to fibrosis stage.

Fibrosis Stage	Total (*n* = 206)		*p*-Value
	*n*	AsAGP (μg/mL)	
Healthy controls	68	1.036 ± 0.308	<0.001
F0-1	39	1.068 ± 0.250	
F2	8	0.9778 ± 0.144	
F3	23	1.261 ± 0.458	
F4 (Liver cirrhosis)	68	1.505 ± 0.432	

AsAGP, asialo α1-acid glycoprotein; F, fibrosis stage.

**Table 4 jcm-12-00712-t004:** Correlation between AsAGP levels, liver stiffness, and other variables.

Variable	AsAGP		Liver Stiffness	
	Coefficient	*p*-Value	Coefficient	*p*-Value
Age (years)	0.075	NS	0.119	0.087
AST (IU/L)	0.227	0.001	0.436	<0.001
ALT (IU/L)	0.160	0.021	0.404	<0.001
Total bilirubin (mg/dL)	0.102	NS	0.139	0.047
Albumin (g/dL)	−0.112	NS	−0.153	0.029
Liver stiffness (kPa)	0.436	<0.001	-	-
AsAGP (μg/mL)	-	-	0.436	<0.001

Note: NS, not significant. AsAGP, asialo α1-acid glycoprotein; ALT, alanine transaminase; AST, aspartate transaminase.

**Table 5 jcm-12-00712-t005:** Diagnostic performance of serum AsAGP levels through the mutual comparison of healthy controls, CHB patients, and CHB patients with LC.

	CHB Patients Versus Healthy Controls	CHB Patients with LC Versus Healthy Controls	CHB Patients Versus CHB Patients with LC
AUC (95% CI)	0.624 (0.528–0.720)	0.821 (0.751–0.890)	0.765 (0.682–0.848)
Optimal cut-off (μg/mL)	0.933	1.036	1.260
Sensitivity	75.7%	83.8%	81.4%
Specificity	55.9%	67.6%	67.5%

AsAGP, asialo α1-acid glycoprotein; CHB, chronic hepatitis B; LC, liver cirrhosis.

**Table 6 jcm-12-00712-t006:** Univariate and multivariate analyses of predictors for detecting CHB with LC.

Variables	Healthy Controls Versus		CHB Patients Versus			
	CHB Patients with LC		CHB Patients with LC			
	Univariate	Stepwise Logistic Regression		Univariate	Stepwise Logistic Regression	
	*p*-Value	OR	*p*-Value	*p*-Value	OR	*p*-Value
Age	0.03			0.341		
Male gender	<0.001			0.629		
Hypertension	-			0.77		
Diabetes	-			0.045		
AST	<0.001			<0.001		
ALT	<0.001	1.17	<0.001	0.011		
GGT	<0.001			0.001	1.012	0.038
Total bilirubin	0.236			0.012		
Albumin	0.013	0.043	0.047	0.019		
Platelet	<0.001	0.97	<0.001	<0.001	0.984	<0.001
AsAGP	<0.001	54.514	<0.001	<0.001	11.898	<0.001

## Data Availability

Data are available on request to the corresponding authors.

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
