# Peer review of "Diagnostic Efficacy of Serum Asialo α1-Acid Glycoprotein Levels for Advanced Liver Fibrosis and Cirrhosis in Patients with Chronic Hepatitis B Compared to That in Healthy Subjects: A Prospective Study"

_jcm, 2023, doi:10.3390/jcm12020712_

Round 1

Reviewer 1 Report

- ABSTRACT/ MANUSCRIPT: you suggest your method is able to predict "liver fibrosis" but your study does not differentiate among fibrosis stages, as suggested in the conclusions; please revise all the text in order to be clearer, thank you.

- LIVER STIFFNESS ASSESSMENT: there is a space after the word "OPERATOR" in the second line; please remove it. 

- You talk about "BASELINE", but what do you suggest as baseline? Please clarify.

- You have healthy controls and patients and, in some cases, you indicate "patients". Please describe all the cohort subjects as "PARTICIPANTS".

- No information is available on the hospital of patients enrolment, please add it. 

- TABLE 6 si not available.

- FUTURE PERSPECTIVES: this is a single centre study. I think you could suggest to analyze different cohorts of patients in future. 

Author Response

We are very grateful for the reviewers` kind and considerate comments on our manuscript entitled “Diagnostic efficacy of serum asialo α1-acid glycoprotein levels for liver fibrosis and cirrhosis in patients with chronic hepatitis B compared to that in healthy subjects: A prospective study” < jcm-2100422 >. We have made a significant effort to make the suggested appropriate changes in accordance with the reviewers’ advice, resulting in a significantly improved manuscript that we hope you will consider for publication in Journal of Clinical Medicine. The revised parts appear in red text in the manuscript.

Sincerely, Prof. Ji Hoon Kim

Department of Internal Medicine, Division of Gastroenterology and Hepatology, Korea University Guro Hospital,

148 Gurodong-ro, Guro-Gu, Seoul 08308, S. Korea

Phone/Fax/Mobile:  +82 2 2626 1030 / +82 2 2626 1038 / +82 10 9089 4395 E-mail: kjhhepar@naver.com

Reviewer 2 Report

Lee et al. present results of a prospective study of the diagnostic efficacy of serum asialo alpha1-acid-glycoprotein levels for liver fibrosis and cirrhosis in patients with chronic hepatitis B compared to that in healthy subjects. 

They found elevated serum AsAGP levels in CHB patients with LC than in healthy controls and suggest a potential role as a biomarker for predicting progressive liver disease in CHB.

The manuscript is well written and visualized. However, I have to raise a few points to be addressed by the authors:

1) Please provide information on the patients' ethnic origin.

2) Noticeable the perecentage of women in the healthy controls was significantly higher than in the CHB groups. Please comment on this.

3) I cannot find Table 6?! Please provide this to retrace the results of your mutivariate analyses.

4) Did you take all blood samples on fasting patients?

5) Regretably, there are no TE information of the healthy control group. In a prospective study I would have definitely expected TE examination in the whole study population as it is a non-invasive cheap assessment. This may bias the results, please comment on this and state this in your limitation section.

Author Response

(The authors gave the same response as above.)
